## META-RESEARCH

# Large-scale language analysis of peer review reports

**Abstract** Peer review is often criticized for being flawed, subjective and biased, but research into peer review has been hindered by a lack of access to peer review reports. Here we report the results of a study in which text-analysis software was used to determine the linguistic characteristics of 472,449 peer review reports. A range of characteristics (including analytical tone, authenticity, clout, three measures of sentiment, and morality) were studied as a function of reviewer recommendation, area of research, type of peer review and reviewer gender. We found that reviewer recommendation had the biggest impact on the linguistic characteristics of reports, and that area of research, type of peer review and reviewer gender had little or no impact. The lack of influence of research area, type of review or reviewer gender on the linguistic characteristics is a sign of the robustness of peer review.

**IVAN BULJAN\*, DANIEL GARCIA-COSTA, FRANCISCO GRIMALDO, FLAMINIO SQUAZZONI AND ANA MARUŠIĆ**

\*For correspondence: ibuljan@ mefst.hr

## Introduction

Most journals rely on peer review to ensure that the papers they publish are of a certain quality, but there are concerns that peer review suffers from a number of shortcomings (*Grimaldo et al., 2018*; *Fyfe et al., 2020*). These include gender bias, and other less obvious forms of bias, such as more favourable reviews for articles with positive findings, articles by authors from prestigious institutions, or articles by authors from the same country as the reviewer (*Haffar et al., 2019*; *Lee et al., 2013*; *Resnik and Elmore, 2016*).

Analysing the linguistic characteristics of written texts, speeches, and audio-visual materials is well established in the humanities and psychology (*Pennebaker, 2017*). A recent example of this is the use of machine learning by Garg et al. to track gender and ethnic stereotypes in the United States over the past 100 years (*Garg et al., 2018*). Similar techniques have been used to analyse scientific articles, with an early study showing that scientific writing is a complex process that is sensitive to formal and informal standards, context-specific canons and subjective factors (*Hartley et al., 2003*). Later studies found that fraudulent scientific papers seem to be less readable than non-fraudulent papers (*Markowitz and Hancock, 2016*), and that papers in economics written by women are better written than equivalent papers by men (and that this gap increases during the peer review process; *Hengel, 2018*). There is clearly scope for these techniques to be used to study other aspects of the research and publishing process.

To date most research on the linguistic characteristics of peer review has focused on comparisons between different types of peer review, and it has been shown that open peer review (in which peer review reports and/or the names of reviewers are made public) leads to longer reports and a more positive emotional tone compared to confidential peer review (*Bravo et al., 2019*; *Bornmann et al., 2012*). Similar techniques have been used to explore possible gender bias in the peer review of grant applications, but a consensus has not been reached yet (*Marsh et al., 2011*; *Magua et al., 2017*). To date, however, these techniques have not been applied to the peer review process at a large scale, largely because most journals strictly limit access to peer review reports.

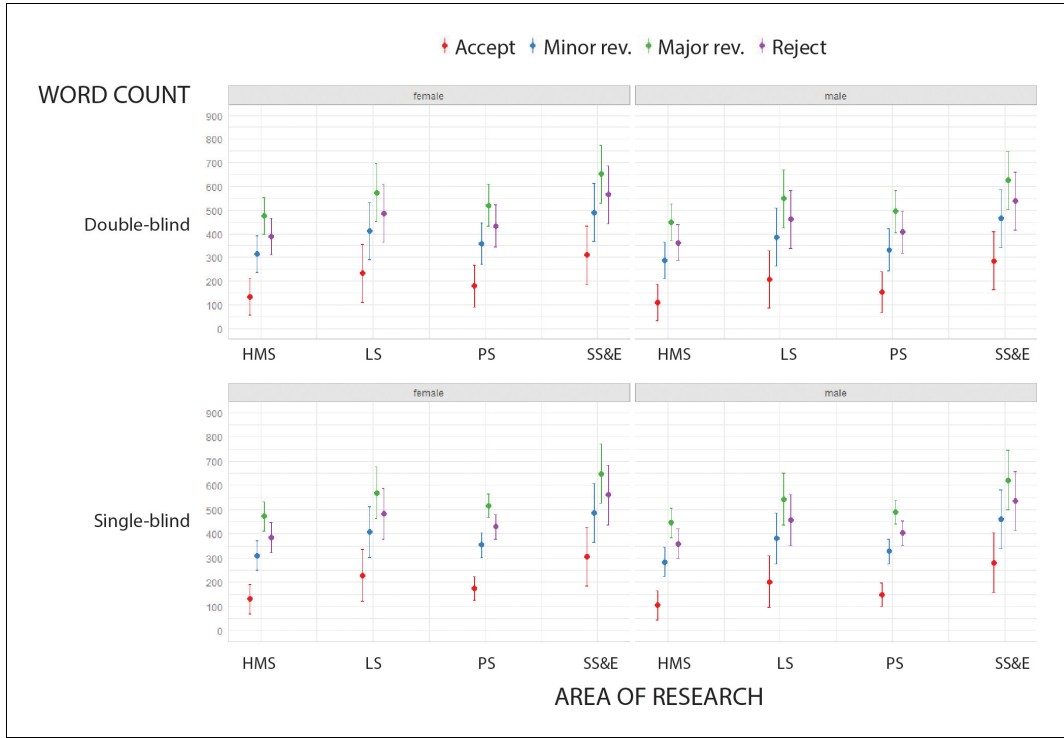

**Figure 1.** Words counts in peer review reports. Word count (mean and 95% confidence interval; LIWC analysis) of peer review reports in four broad areas of research for double-blind review (top) and single-blind review (bottom), and for female reviewers (left) and male reviewers (right). Reports recommending accept (red) were consistently the shortest, and reports recommending major revisions (green) were consistently the longest. See *Supplementary file 1* for summary data and mixed model linear regression coefficients and residuals. HMS: health and medical sciences; LS: life sciences; PS: physical sciences; SS&E: social sciences and economics.

Here we report the results of a linguistic analysis of 472,449 peer review reports from the PEERE database (*Squazzoni et al., 2017*). The reports came from 61 journals published by Elsevier in four broad areas of research: health and medical sciences (22 journals); life sciences (5); physical sciences (30); social sciences and economics (4). For each review we had data on the following: i) the recommendation made by the reviewer (accept [n = 26,387, 5.6%]; minor revisions required [134,858, 28.5%]; major revisions required [161,696, 34.2%]; reject [n = 149,508, 31.7%]); ii) the broad area of research; iii) the type of peer review used by the journal (single-blind [n = 411,727, 87.1%] or double-blind [n = 60,722, 12.9%]); and the gender of the reviewer (75.9% were male; 24.1% were female).

## Results

We used various linguistic tools to examine the peer review reports in our sample (see Methods for more details). Linguistic Inquiry and Word Count (LIWC) text-analysis software was used to perform word counts and to return scores of between 0% and 100% for 'analytical tone', 'clout' and 'authenticity' (*Pennebaker et al., 2015*). Three different approaches were used to perform sentiment analysis: i) LIWC returns a score between 0% and 100% for 'emotional tone' (with more positive emotions leading to higher scores); ii) the SentimentR package returns a majority of scores between –1 (negative sentiment) and +1 (positive sentiment), with an extremely low number of results outside that range (0.03% in our sample); iii) the Stanford CoreNLP returns a score between 0 (negative sentiment) to +4 (positive sentiment). We also used LIWC to analyse the reports in terms of five foundations of morality (*Graham et al., 2009*).

### Length of report

For all combinations of area of research, type of peer review and reviewer gender, reports recommending accept were shortest, followed by reports recommending minor revisions, reject, and major revisions (*Figure 1*). Reports written by reviewers for social sciences and economics

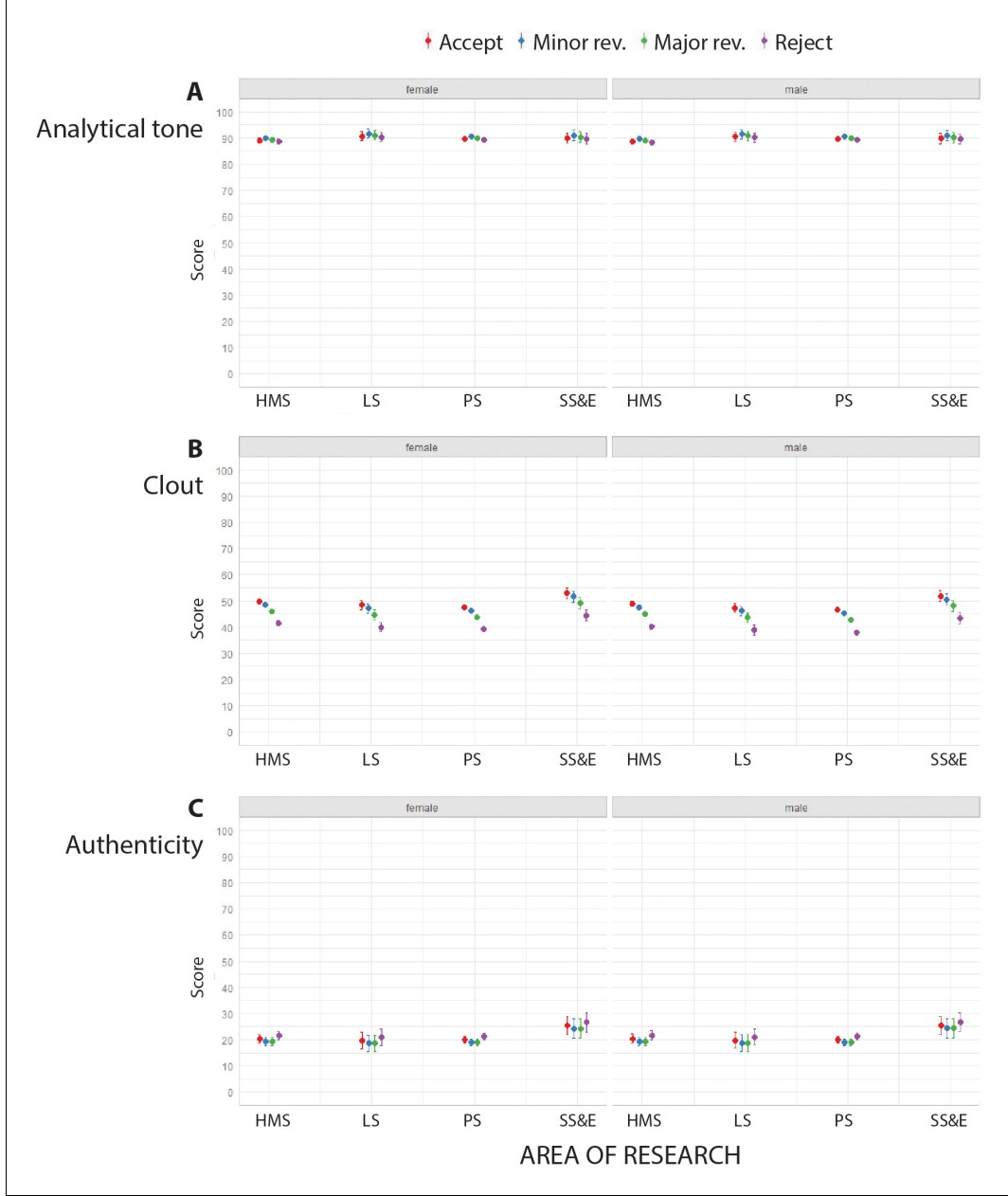

**Figure 2.** Analytical tone, clout and authenticity and in peer review reports for single-blind review. Scores returned by LIWC (mean percentages and 95% confidence interval) for analytical tone (**A**), clout (**B**) and authenticity (**C**) for peer review reports in four broad areas of research for female reviewers (left) and male reviewers (right) using single-blind review. Reports recommending accept (red) consistently had the most clout, and reports recommending reject (purple) consistently had the least clout. See *Supplementary files 2–4* for summary data, mixed model linear regression coefficients and residuals, and examples of reports with high and low scores for analytical tone, clout and authenticity. HMS: health and medical sciences; LS: life sciences; PS: physical sciences; SS&E: social sciences and economics.

The online version of this article includes the following figure supplement(s) for figure 2:

**Figure supplement 1.** Analytical tone, clout and authenticity in peer review reports for double-blind review.

journals were significantly longer than those written by reviewers for medical journals; men also tended to write longer reports than women; however, the type of peer review (i.e., single- vs. double-blind) did not have any influence on the length of reports (see Table 2 in *Supplementary file 1*).

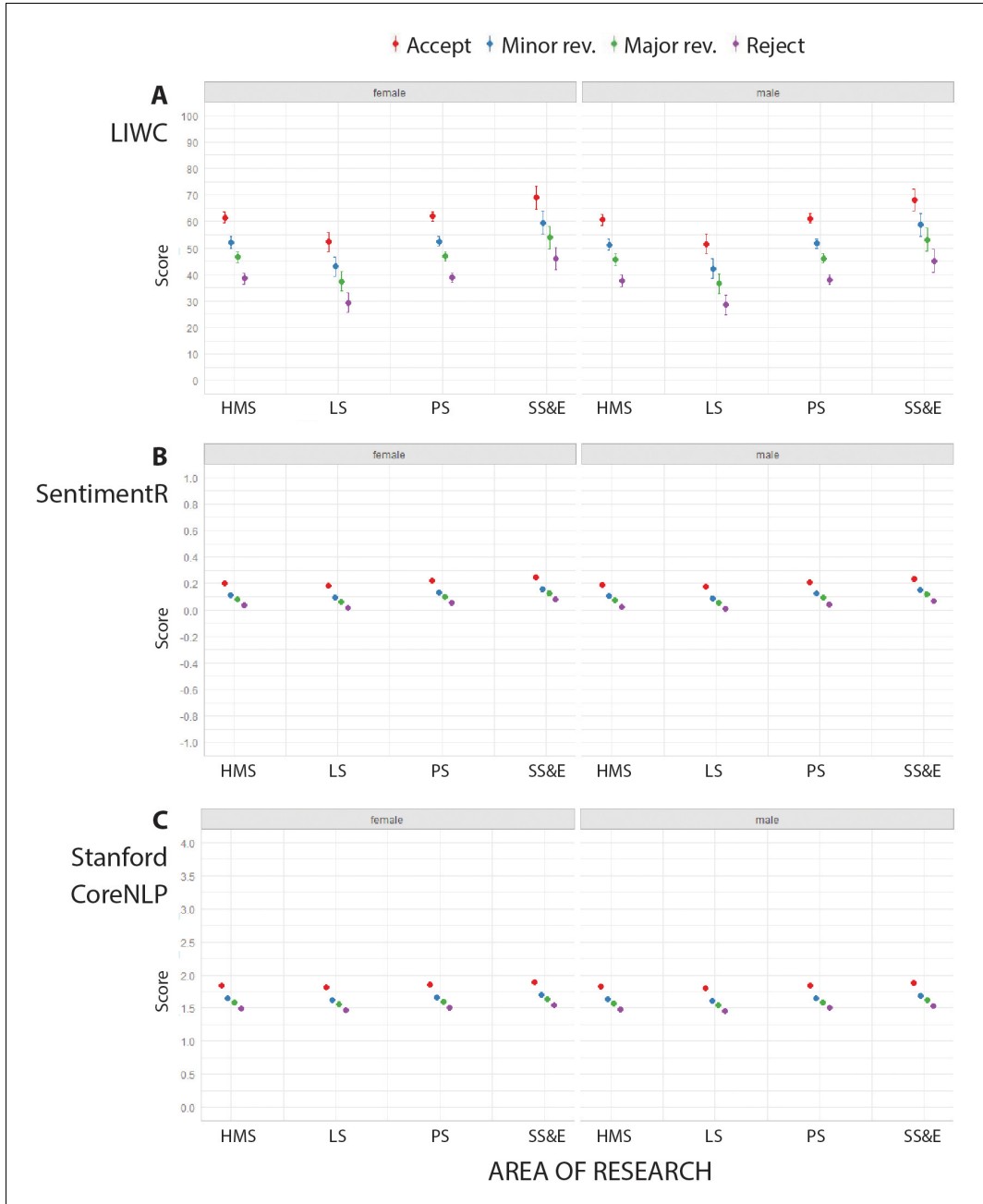

**Figure 3.** Sentiment analysis of peer review reports for single-blind review. Scores for sentiment analysis returned by LIWC (**A**; mean percentage and 95% confidence interval, CI), SentimentR (**B**; mean score and 95% CI), and Stanford CoreNLP (**C**; mean score and 95% CI) for peer review reports in four broad areas of research for female reviewers (left) and male reviewers (right) using single-blind review. See **Supplementary files 5–7** for summary data, mixed model linear regression coefficients and residuals, and examples of reports with high and low scores for sentiment according to LIWC, SentimentR and Stanford CoreNLP analysis.
The online version of this article includes the following figure supplement(s) for figure 3:

**Figure supplement 1.** Sentiment analysis of peer review reports for double-blind review.

### Analytical tone, clout and authenticity

LIWC returned high scores (typically between 85.0 and 91.0) for analytical tone, and low scores (typically between 18.0 and 25.0) for authenticity, for the peer review reports in our sample (*Figure 2A,C*; *Figure 2—figure supplement 1A,C*). High authenticity of a text is defined as the use of more personal words (I-

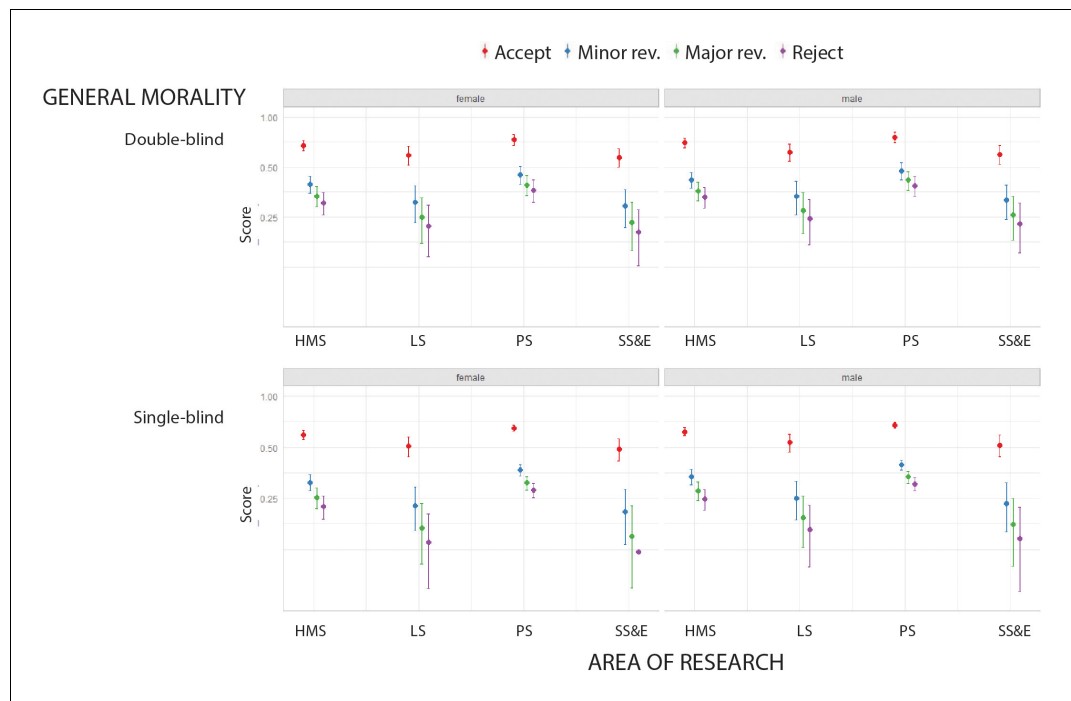

**Figure 4.** Moral foundations in peer review reports. Scores returned by LIWC (mean percentage on a log scale) for general morality in peer review reports in four broad areas of research for double-blind review (top) and single-blind review (bottom), and for female reviewers (left) and male reviewers (right). Reports recommending accept (red) consistently had the highest scores. See *Supplementary file 8* for lists of the ten most frequent words found in peer review reports for general morality and the five moral foundation variables. HMS: health and medical sciences; LS: life sciences; PS: physical sciences; SS&E: social sciences and economics.

The online version of this article includes the following figure supplement(s) for figure 4:

**Figure supplement 1.** Scores returned by LIWC (mean percentage on a log scale and 95% CI) for care/harm, one of the five foundations of Moral Foundations Theory.
**Figure supplement 2.** Scores returned by LIWC (mean percentage on a log scale and 95% CI) for fairness/cheating, one of the five foundations of Moral Foundations Theory.
**Figure supplement 3.** Scores returned by LIWC (mean percentage on a log scale and 95% CI) for loyalty/betrayal, one of the five foundations of Moral Foundations Theory.
**Figure supplement 4.** Scores returned by LIWC (mean percentage on a log scale and 95% CI) for authority/subversion, one of the five foundations of Moral Foundations Theory.
**Figure supplement 5.** Scores returned by LIWC (mean percentage on a log scale and 95% CI) for sanctity/degradation, one of the five foundations of Moral Foundations Theory.

words), present tense words, and relativity words, and fewer non-personal words and modal words (*Pennebaker et al., 2015*). Low authenticity and high analytical tone are characteristic of texts describing medical research (*Karačić et al., 2019*; *Glonti et al., 2017*). There was some variation with reviewer recommendation in the scores returned for clout, with accept having the highest scores for clout, followed by minor revisions, major revisions and reject (*Figure 2B*; *Figure 2—figure supplement 1B*).

When reviewers recommended major revisions, the text of the report was more analytical. The analytical tone was higher when reviewers were women and for single-blind peer review,

but we did not find any effect of the area of research (see Table 4 in *Supplementary file 2*).

Clout levels varied with area of research, with the highest levels in social sciences and economics journals (see Table 7 in *Supplementary file 3*). When reviewers recommended rejection, the text showed low levels of clout, as it did when reviewers were men and when the journal useded single-blind peer review (see Table 7 in *Supplementary file 3*).

The text of reports in social sciences and economics journals had the highest levels of authenticity. Authenticity was prevalent also when reviewers recommended rejection. There was no significant variation in terms of authenticity per

reviewer gender or type of peer review (see Table 10 in *Supplementary file 4*).

### Sentiment analysis

The three approaches were used to perform sentiment analysis on our sample – LIWC, SentimentR and the Stanford CoreNLP – produced similar results. Reports recommending accept had the highest scores, indicating higher sentiment, followed by reports recommending minor revisions, major revisions and reject (*Figure 3*; *Figure 3—figure supplement 1*). Furthermore, reports for social sciences and economics journals had the highest levels of sentiment, as did reviews written by women. We did not find any association between sentiment and the type of peer review (see Table 13 in *Supplementary file 5*, Table 16 in *Supplementary file 6* and Table 19 in *Supplementary file 7*).

### Moral foundations

LIWC was also used to explore the morality of the reports in our sample (*Graham et al., 2009*). The differences between peer review recommendations were statistically significant. Reports recommending acceptance had the highest scores for general morality, followed by reports recommending minor revisions, major revisions and reject (*Figure 4*). Regarding the research area, we found a lowest proportion of words related to morality in the social sciences and economics, when reviewers were men, and when single-blind peer review was used (*Figure 4*).

We also explored five foundations of morality – care/harm, fairness/cheating, loyalty/betrayal, authority/subversion, and sanctity/degradation – but no clear patterns emerged (*Figure 4—figure supplements 1–5*). See the Methods section for more details, and *Supplementary file 8* for lists of the ten most common phrases from the LIWC Moral Foundation dictionary. In general, the prevalence of these words was minimal, with average scores lower than 1%. Moreover, these words tended to be part of common phrases and thus did not speak to the moral content of the reviews. This suggests that a combination of qualitative and quantitative methods, including machine learning tools, will be required to explore the moral aspects of peer review.

## Conclusion

Our study suggests that the reviewer recommendation has the biggest influence on the linguistic characteristics (and length) of peer review reports, which is consistent with previous, case-based research (*Casnici et al., 2017*). It is probable that whenever reviewers recommend revision, they write a longer report in order to justify their requests and/or to suggest changes to improve the manuscript (which they do not have to do when they recommend to accept or reject). In our study, in the case of the two more negative recommendations (reject and major revisions), the reports were shorter, and language was less emotional and more analytical. We found that the type of peer review – single-blind or double-blind – had no significant influence on the reports, contrary to previous reports on smaller samples (*Bravo et al., 2019*; *van Rooyen et al., 1999*). Likewise, area of research had no significant influence on the reports in the sample, and neither did reviewer gender, which is consistent with a previous smaller study (*Bravo et al., 2019*). The lack of influence exerted by the area of research, the type of peer review or the reviewer gender on the linguistic characteristics of the reports is a sign of the robustness of peer review.

The results of our study should be considered in the light of certain limitations. Most of the journals were in the health and medical sciences and the physical sciences, and most used single-blind peer review. However, the size, depth and uniqueness of our dataset helped us provide a more comprehensive analysis of peer review reports than previous studies, which were often limited to small samples and incomplete data (*van den Besselaar et al., 2018*; *Sizo et al., 2019*; *Falk Delgado et al., 2019*). Future research would also benefit from baseline data against which results could be compared, although our results match the preliminary results from a study at a single biomedical journal (*Glonti et al., 2017*), and from knowing more about the referees (such as their status or expertise). Finally, we did not examine the actual content of the manuscripts under review, so we could not determine how reliable reviewers were in their assessments. Combining language analyses of peer review reports with estimates of peer review reliability for the same manuscripts (via inter-reviewer ratings) could provide new insights into the peer review process.

## Methods

### The PEERE dataset

PEERE is a collaboration between publishers and researchers (*Squazzoni et al., 2020*), and the PEERE dataset contains 583,365 peer review

reports from 61 journal published by Elsevier, with data on reviewer recommendation, area of research (health and medical sciences; life sciences; physical sciences; social sciences and economics), type of peer review (single blind or double blind), and reviewer gender for each report. Most of the reports (N = 481,961) are for original research papers, with the rest (N = 101,404) being for opinion pieces, editorials and letters to the editor. The database was first filtered to exclude reviews that included reference to manuscript revisions, resulting in 583,365 reports. We eliminated 110,636 due to the impossibility to determine reviewer gender, and 260 because we did not have data on the recommendation. Our analysis was performed on a total number of 472,449 peer review reports.

### Gender determination

To determine reviewer gender, we followed a standard disambiguation algorithm that has already been validated on a dataset of scientists extracted from the Web of Science database covering a similar publication time window (*Santamaría and Mihaljević, 2018*). Gender was assigned following a multi-stage gender inference procedure consisting of three steps. First, we performed a preliminary gender determination using, when available, gender salutation (i. e., Mr, Mrs, Ms...). Secondly, we queried the Python package gender-guesser about the extracted first names and country of origin, if any. Gender-guesser has demonstrated to achieve the lowest misclassification rate and introduce the smallest gender bias (*Paltridge, 2017*). Lastly, we queried the best performer gender inference service, Gender API (https://gender-api.com/), and used the returned gender whenever we found a minimum of 62 samples with, at least, 57% accuracy, which follows the optimal values found in benchmark 2 of the previous research (*Santamaría and Mihaljević, 2018*). This threshold for the obtained confidence parameters was suitable to ensure that the rate of misclassified names did not exceed 5% (*Santamaría and Mihaljević, 2018*). This allowed us to determine the gender of 81.1% of reviewers, among which 75.9% were male and 24.1% female. With regards to the three possible gender sources, 6.3% of genders came from scientist salutation, 77.2% from gender-guesser, and 16.5% from the Gender API. The remaining 18.9% of reviewers were assigned an unknown gender. This level of gender determination is consistent with the non-classification rate for names of scientists in previous research (*Santamaría and Mihaljević, 2018*).

### Analytical tone, authenticity and clout

We used a version of the Linguistic Inquiry and Word Count (LIWC) text-analysis software with standardized scores (http://liwc.wpengine.com/) to analyze the peer review reports in our sample. LIWC measures the percentage of words related to three psychological features (so scores range from 0 to 100): 'analytical tone'; 'clout'; and "authenticity. A high score for analytical tone indicates a report with a logical and hierarchical style of writing. Clout reveals personal sensitivity towards social status, confidence or leadership: a low score for clout is associated with insecurities and a less confident and more tentative tone (*Kacewicz et al., 2014*). A high score for authenticity indicates a report written in a style that is honest and humble, whereas a low score indicates a style that is deceptive and superficial (*Pennebaker et al., 2015*). The words people use also reflect how authentic or personal they sound. People who are authentic tend to use more I-words (e.g. I, me, mine), present-tense verbs, and relativity words (e.g. near, new) and fewer she-he words (e.g. his, her) and discrepancies (e.g. should, could) (*Pennebaker et al., 2015*).

### Sentiment analysis

We used three different methodological approaches to assess sentiment. (i) LIWC measures 'emotional tone', which indicates writing dominated by either positive or negative emotions by counting number of words from a pre-specified dictionary. (ii) The SentimentR package (*Rinker, 2019*) classifies the proportion of words related to sentiment in the text, similarly to the 'emotional tone' scores in LIWC but using a different vocabulary. The SentimentR score is the valence of words related with the specific sentiment, majority of scores (99.97%) ranging from −1 (negative sentiment) +1 (positive sentiment). (iii) Stanford CoreNLP is a deep language analysis program that uses machine learning to determine the emotional valence of the text (*Socher et al., 2013*), and score ranges from 0 (negative sentiment) to +4 (positive sentiment). Examples of characteristic text variables from the peer review reports analysed with these approaches are given in *Supplementary files 5–7*.

### Moral foundations

We used LIWC and Moral Foundations Theory (https://moralfoundations.org/other-materials/) to analyse the reports in our sample according to five moral foundations: care/harm (also known as care-virtue/care-vice); fairness/cheating (or fairness-virtue/fairness-vice); loyalty/betrayal (or loyalty-virtue/loyalty-vice); authority/subversion (authority virtue/authority-vice); and sanctity/degradation (or sanctity-virtue/sanctity-vice).

### Statistical methods

Data were analysed using the R programming language, version 3.6.3. (*R Development Core Team, 2017*). To test the interaction effects and compare different peer review characteristics, we conducted a mixed model linear analysis on each variable (analytical tone, authenticity, clout; the measures of sentiment; and the measures of morality) with reviewer recommendation, area of research, type of peer review (single- or double-blind) and reviewer gender as fixed factors (predictors) and the journal, word count and article type as the random factor. This was to control across-journal interactions, number of words and article type.

### Acknowledgements

We thank Dr Bahar Mehmani from Elsevier for helping us with data collection.

**Ivan Buljan** is in the Department of Research in Biomedicine and Health, University of Split School of Medicine, Split, Croatia
ibuljan@mefst.hr
https://orcid.org/0000-0002-8719-7277

**Daniel Garcia-Costa** is in the Department d'Informàtica, University of Valencia, Burjassot-València, Spain
https://orcid.org/0000-0002-8939-8451

**Francisco Grimaldo** is in the Department d'Informàtica, University of Valencia, Burjassot-València, Spain
https://orcid.org/0000-0002-1357-7170

**Flaminio Squazzoni** is in the Department of Social and Political Sciences, University of Milan, Milan, Italy
https://orcid.org/0000-0002-6503-6077

**Ana Marušić** is in the Department of Research in Biomedicine and Health, University of Split School of Medicine, Split, Croatia
https://orcid.org/0000-0001-6272-0917

*Author contributions:* Ivan Buljan, Conceptualization, Data curation, Formal analysis, Investigation, Visualization, Methodology, Writing - original draft, Writing - review and editing; Daniel Garcia-Costa, Data curation, Software, Formal analysis, Investigation, Visualization, Writing - original draft, Writing - review and editing; Francisco Grimaldo, Conceptualization, Data curation, Software, Formal analysis, Funding acquisition, Investigation, Visualization, Methodology, Writing - original draft, Writing - review and editing; Flaminio Squazzoni, Conceptualization, Resources, Data curation, Supervision, Investigation, Methodology, Writing - original draft, Project administration, Writing - review and editing; Ana Marušić, Conceptualization, Resources, Supervision, Funding acquisition, Methodology, Writing - original draft, Project administration

**Funding**

| Funder | Grant reference number | Author |
|---|---|---|
| Ministerio de Ciencia e Innovación | RTI2018-095820-B-I00 | Daniel Garcia-Costa Francisco Grimaldo |
| Spanish Agencia Estatal de Investigación | RTI2018-095820-B-I00 | Daniel Garcia-Costa Francisco Grimaldo |
| European Regional Development Fund | RTI2018-095820-B-I00 | Daniel Garcia-Costa Francisco Grimaldo |
| Croatian Science Foundation | IP-2019-04-4882 | Ana Marušić |

The funders had no role in study design, data collection and interpretation, or the decision to submit the work for publication.

**Decision letter and Author response**
Decision letter https://doi.org/10.7554/eLife.53249.sa1
Author response https://doi.org/10.7554/eLife.53249.sa2

## Additional files

### Supplementary files

• Supplementary file 1. Word count (*Figure 1*): summary data and mixed model linear regression coefficients and residuals.

• Supplementary file 2. Analytical tone (*Figure 2A*): summary data, mixed model linear regression coefficients and residuals, and examples of reports with high and low scores for LIWC analytical tone.

• Supplementary file 3. Clout (*Figure 2B*): summary data, mixed model linear regression coefficients and residuals, and examples of reports with high and low scores for LIWC clout.

• Supplementary file 4. Authenticity (*Figure 2C*): summary data, mixed model linear regression coefficients and residuals, and examples of reports with high and low scores for LIWC authenticity.

- Supplementary file 5. Sentiment/LIWC emotional tone (*Figure 3A*): summary data, mixed model linear regression coefficients and residuals, and examples of reports with high and low scores for sentiment (LIWC emotional tone).

- Supplementary file 6. Sentiment/SentimentR score (*Figure 3B*): summary data, mixed model linear regression coefficients and residuals, and examples of reports with high and low scores for sentiment (SentimentR scores).

- Supplementary file 7. Sentiment/Stanford CoreNLP score (*Figure 3C*): summary data, mixed model linear regression coefficients and residuals, and examples of reports with high and low scores for sentiment (Stanford CoreNLP score).

- Supplementary file 8. Ten most frequent words found in peer review reports for general morality and the five moral foundation variables.

- Transparent reporting form

## Data availability

The journal dataset required a data sharing agreement to be established between authors and publishers. A protocol on data sharing entitled 'TD1306 COST Action New frontiers of peer review (PEERE) PEERE policy on data sharing on peer review' was signed by all partners involved in this research on 1 March 2017, as part of a collaborative project funded by the EU Commission. The protocol established rules and practices for data sharing from a sample of scholarly journals, which included a specific data management policy, including data minimization, retention and storage, privacy impact assessment, anonymization, and dissemination. The protocol required that data access and use were restricted to the authors of this manuscript and data aggregation and report were done in such a way to avoid any identification of publishers, journals or individual records involved. The protocol was written to protect the interests of any stakeholder involved, including publishers, journal editors and academic scholars, who could be potentially acted by data sharing, use and release. The full version of the protocol is available on the peere.org website. To request additional information on the dataset and for any claim or objection, please contact the PEERE data controller at info@peere.org.

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
