## [Decision Letter]

Thank you for submitting your article "Meta research: Large-scale language analysis of peer review reports reveals lack of moral bias" to *eLife* for consideration as a Feature Article. Your article has been reviewed by three peer reviewers, and the evaluation has been overseen by the *eLife* Features Editor. One of the reviewers was Erin Hengel; the other two reviewers have opted to remain anonymous.

The reviewers have discussed the reviews with one another and the Reviewing Editor has drafted this decision to help you prepare a revised submission.

Summary:

The paper presents an explorative study of the linguistic content of a very large set (roughly 500,000) of peer review reports. To our knowledge, this is the first paper addressing such a large data set on peer review and the potential biases often experienced by scholars of all fields. This is highly valuable to academic research as a whole. However, we do not believe the evidence warrants the authors' conclusion that peer review lacks moral bias (see below), and we suggest that the authors revise their paper to focus exclusively on the LIWC indicators (and conduct further research into moral bias, informed by the comments from referee #2 at the end of this email, with a view to writing a second manuscript for future submission). There are also a number of other issues that need to be addressed in a revised manuscript.

Essential revisions:

1. Reducing the emphasis on moral bias in the present manuscript will involve deleting the last 10 rows of table 1 (and similarly for tables 2, 3 and 4), and making changes to the text. The title also needs to be changed, but Figure 1 does not need to be changed.

2. The authors should analyze the data they have for word length, analytical thinking, clout, authenticity, emotional tone and morality in general in greater detail through regression analyses and/or multilevel models.

Here is what referee #1 said on this topic: I am concerned that the explorative approach misses a lot of underlying information. By comparing means of groups, the in-group variance is overlooked. Some of the observed effects might be large enough to hold some sort of universal "truth", but for other cases, substantial effects might still exist within groups. My suggestion would be to rethink the design into one that allows for multidimensionality. One approach could be regression analysis, which would require a more strict type of test-design though. Another approach could be to build reviewer-profiles based on their characteristics. E.g. reviews with a high degree of clout and negative emotion, and low analytical thinking could be one type. Where is this type found? What characterizes it? Additionally, this would allow the authors to include information about the reviewer, e.g. are they always negative? This might also solve the baseline problem, as this would be a classification issue rather than a measurement issue.

Here is what referee #3 said on this topic: All the analyses are descriptive ANOVAs and the like (bivariate). Given the supposed quality of the data, I'd recommend they explore multilevel models. Then they can fix out differences by fields, journals, etc so we get a more clear sense of what's going on.

3. Please include a fuller description of your dataset and describe how representative/biased the sampling is by field and type of journal.

4. Please provide more information on what the LIWC measures and how it is calculated? It would be especially helpful if you showed several LIWC scores together with sample texts (preferably from your own database) to illustrate how well it analyses text. This can be shown in an appendix. If you can't show text from your own database due to privacy concerns, feel free to show passages from this report. (Alternatively, you could take a few reports from, say, the BMJ which makes referee reports publicly available for published papers.)

5. What are baseline effects? What kind of changes should we expect? E.g. when arguing that the language is cold and analytical, what is this comparable to? I would expect most scientific writing to be mostly in this category, and it should not be a surprise - I hope. It would be very useful for the reader to have some type of comparison.

6. Does the analysis allow for the fact that different subject areas will have different LIWC scores? The largest sample of reports comes from the physical sciences, which use single-blind review the most. Reports from this field are also shorter, slightly more analytic and display less Clout and Authenticity. I think your results are picking up this selection bias instead of representing actual differences between the two review processes.

Please discuss.

7. Please add a section on the limitations of your study. For example, there is no discussion of sample bias and representation really.

Also, LIWC is full of issues and problems (NLP has come a ways since LIWC arrived on the scene): do you use the standardized version of LIWC constructs with high cronbach alphas, or the raw versions with poor cronbach alphas?

Does the analysis distinguish between original research content and other forms of content (eg, editorials, opinion pieces, book reviews etc)?

---

## [Author Response]

[We repeat the reviewers’ points here in italic, and include our replies point by point in Roman.]

Essential revisions:1. Reducing the emphasis on moral bias in the present manuscript will involve deleting the last 10 rows of table 1 (and similarly for tables 2, 3 and 4), and making changes to the text. The title also needs to be changed, but Figure 1 does not need to be changed.

Thank you for your comment. We revised the tables according to the reviewers’ recommendations and created new tables where we present multilevel description of the review reports based on the interaction effects of the reviewer recommendation, journal discipline, journal peer review type and reviewer gender. We created the tables for the five LIWC variables (word count, analytical tone, clout tone, authenticity and emotional tone), and two new additional measures (SentimentR- an R package that has its own dictionaries for the emotional tone of the text, and Stanford CoreNLP-a deep language analysis software, which served as the concurrent validity assessment of the tone variables from the LIWC package). All multilevel relations are now presented in new figures, which are the results of the mixed methods linear regression where we controlled the random effect of the journal, word count (except for LIWC word count) and article type. In light of the new results, and coherently with referee recommendations, we introduced a change in the title and replaced the Figure 1. We now have seven new graphs describing linguistic characteristics of the reviews between groups in the main text and eleven graphs presenting the moral variables in the Supplementary file.

2. The authors should analyze the data they have for word length, analytical thinking, clout, authenticity, emotional tone and morality in general in greater detail through regression analyses and/or multilevel models.

Thank you for your comment. As mentioned above, we performed a new analysis, where we used the mixed methods approach with reviewer recommendation, journal discipline, journal peer review type and reviewer gender as predictors (fixed factors) and different journals, word count and article type as the random factor, which would enable us to control the variations between journals (there were 61 journals in total from which some were more represented than others, and the majority of the articles were original research articles). We found significant interaction of the reviewer recommendation, journal’s field of research, the type of peer review and reviewer’s gender in each variable assessment, but we understand that this significance could be due to the large sample size. So, we presented figures with the within-group relations on standardized scales where we presented the differences between groups.

Here is what referee #1 said on this topic: I am concerned that the explorative approach misses a lot of underlying information. By comparing means of groups, the in-group variance is overlooked. Some of the observed effects might be large enough to hold some sort of universal "truth", but for other cases, substantial effects might still exist within groups. My suggestion would be to rethink the design into one that allows for multidimensionality. One approach could be regression analysis, which would require a stricter type of test-design though. Another approach could be to build reviewer-profiles based on their characteristics. E.g. reviews with a high degree of clout and negative emotion, and low analytical thinking could be one type. Where is this type found? What characterizes it? Additionally, this would allow the authors to include information about the reviewer, e.g. are they always negative? This might also solve the baseline problem, as this would be a classification issue rather than a measurement issue.

Thank you for the comment. Excellent point. We re-performed the analysis accordingly. The mixed methods approach revealed that the majority of effects of the differences in the writing style of the reviews can be attributed to reviewer recommendations, much less to the journal’s field of research, the type of peer review type and reviewer’s gender. We tried to provide an overview of the general writing style in peer reviews by presenting relevant variables in the same graphs so that a reader can have an overview about what peer review characteristics predict different language styles.

Here is what referee #3 said on this topic: All the analyses are descriptive ANOVAs and the like (bivariate). Given the supposed quality of the data, I'd recommend they explore multilevel models. Then they can fix out differences by fields, journals, etc so we get a clearer sense of what's going on.

As explained above, we performed mixed methods approach where these effects were analysed jointly. The current analysis provides an overview of the interaction effects in peer review characteristics and sizes of the differences between them.

3. Please include a fuller description of your dataset and describe how representative/biased the sampling is by field and type of journal.

Thank you, we added this both to the methods and the limitations in the Discussion section.

4. Please provide more information on what the LIWC measures and how it is calculated? It would be especially helpful if you showed several LIWC scores together with sample texts (preferably from your own database) to illustrate how well it analyses text. This can be shown in an appendix. If you can't show text from your own database due to privacy concerns, feel free to show passages from this report. (Alternatively, you could take a few reports from, say, the BMJ which makes referee reports publicly available for published papers.)

LIWC has a dictionary with words associated with different tone, and it counts number of words for each tone type in a certain text. The LIWC output is the percentage of words from a tone category in the text. We now provided the calculation of the different tone variables in the Supplementary file, both for high and low levels of tone. The examples are anonymized.

5. What are baseline effects? What kind of changes should we expect? E.g. when arguing that the language is cold and analytical, what is this comparable to? I would expect most scientific writing to be mostly in this category, and it should not be a surprise - I hope. It would be very useful for the reader to have some type of comparison.

Another good point. The results in our study were similar to an unpublished study that focused on the analysis of the peer review linguistic characteristics, but on a much smaller sample and in only in a single journal (https://peerreviewcongress.org/prc17-0234). However, with the new methodological approach we looked at the relationship of the linguistic characteristics and different aspects of peer review process and found important differences. The analytical tone was indeed predominant in all types of peer review reports, but we found differences in other linguistic characteristics. The new results are presented in the revised manuscript.

6. Does the analysis allow for the fact that different subject areas will have different LIWC scores? The largest sample of reports comes from the physical sciences, which use single-blind review the most. Reports from this field are also shorter, slightly more analytic and display less Clout and Authenticity. I think your results are picking up this selection bias instead of representing actual differences between the two review processes.

Thank you for your comment. The dataset characteristics are now described in the limitations in the Discussion section and we are aware that there is a higher prevalence of journals from Physical sciences, double blind reviews and those which asked for revisions. However, the new analyses now include the interaction of peer review characteristics and so we introduced a better control for this selection bias.

7. Please add a section on the limitations of your study. For example, there is no discussion of sample bias and representation really.

As mentioned previously, we added a limitation section to the revised manuscript.

Also, LIWC is full of issues and problems (NLP has come a ways since LIWC arrived on the scene): do you use the standardized version of LIWC constructs with high cronbach alphas, or the raw versions with poor cronbach alphas?

The LIWC version we used is the standardized version with high Cronbach alphas. This has now been clarified in the Methods section of the revised manuscript. We also analysed the data using Stanford CoreNLP deep learning tool in order to increase internal validity of our approach.

Does the analysis distinguish between original research content and other forms of content (eg, editorials, opinion pieces, book reviews etc)?

There were no book reviews in the dataset. However, we did make the distinction between the original articles and other formats (There was the total of 388,737 original articles and 83,972 of other types or articles of those included in the mixed model analyses), which is now described in the Methods. We used this as the random factor in the mixed model linear regression.